# Microclimatic Fluctuation throughout the Day Influences Subtropical Fruit-Feeding Butterfly Assemblages between the Canopy and Understory

Aline Richter [1,2,*], Milton de Souza Mendonça, Jr. [2], Karine Gawlinski [2] and Cristiano Agra Iserhard [1,*]

[1] Programa de Pós-Graduação em Biodiversidade Animal, Departamento de Ecologia, Zoologia e Genética, Universidade Federal de Pelotas, Pelotas 96.160-000, Rio Grande do Sul, Brazil

[2] Programa de Pós-Graduação em Ecologia, Departamento de Ecologia, Universidade Federal do Rio Grande do Sul, Porto Alegre 91.509-900, Rio Grande do Sul, Brazil; milton.mendonca@ufrgs.br (M.d.S.M.J.); kah_g@hotmail.com (K.G.)

* Correspondence: linebio.r@gmail.com (A.R.); cristianoagra@yahoo.com.br (C.A.I.)

**Abstract:** Vertical stratification is a recognized pattern in tropical forests; however, biotic and abiotic factors driving this pattern are little explored. We investigated the influence of daily climatic variation in the vertical stratification of fruit-feeding butterfly assemblages sampled with bait traps in the understory and canopy of a subtropical Atlantic Forest. Overall, 1347 individuals belonging to 38 species of fruit-feeding butterflies were recorded. The canopy and understory are distinct concerning diurnal but not nocturnal microclimatic conditions, leading to different responses in community structure. Richness did not differ between strata, but we observed an effect of stratum in interaction with variation in microclimatic conditions, with the canopy increasing in abundance compared to the understory. Temperature homogenization at night can hinder vertical stratification in richness, while microclimatic variation influences species abundance. The species composition was affected by strata with high turnover in the understory, without an effect of microclimatic variables in beta diversity. In addition to the difference in composition, our study shows that the understory was represented mainly by species from Satyrinae, while the canopy presented species from different clades. This could be an artefact of habitat structure, and the species adapted to the closed forest have a dispersal limitation compared to in the canopy. These findings help us to better understand the mechanisms generating distinct patterns of vertical stratification of fruit-feeding butterflies in the Neotropics and provide new insights into the role of microclimatic conditions in the structure of insect assemblages.

**Keywords:** araucaria forest; beta diversity; community ecology; humidity; Nymphalidae; temperature; vertical stratification





## 1. Introduction

Tropical and subtropical forests support a high insect diversity [1,2], and a large portion of this diversity seems to be associated with the canopy [3]. Forest vertical stratification in distinct layers, such as the understory and canopy, leads to variations in microhabitat conditions at the local scale (e.g., light intensity, temperature, and wind speed, leading to differences in resource availability for different organisms) and, hence, can affect invertebrate distribution [3–5]. Subtropical forests, located at latitudes between 24° and 34° in the Northern and Southern hemispheres, are generally less structurally complex, have lower plant species richness, and have shorter canopy heights than tropical rainforests [3,6]. These characteristics could lead to less pronounced vertical stratification, causing a reduction in the alpha diversity of canopy-specialized organisms while maintaining a high disparity between strata (and, thus, potentially high beta diversity) [7–10]. Furthermore, other subtropical forest features could maintain a vertical stratification, such as daily climatic

fluctuation [2,11], promoting relatively high differences in temperature during the day but homogenizing temperature at night [12]. In addition to the effects of climate and forest height, forests in elevated areas can exhibit further variations in temperature, light availability, and precipitation, consequently affecting the distribution of insects [13].

The Atlantic Forest is a heterogeneous and threatened Brazilian biome considered a diversity hotspot with high rates of endemism, including butterfly species [1,14,15]. The Araucaria Forest belongs to the Atlantic Forest domain and is viewed as a transitional zone between dense ombrophilous forest and seasonal semideciduous forest, forming a conspicuous environmental transition that leads to high species turnover [16]. The Araucaria forests extend across the south of Brazil between latitudes 20° S and 29° S and are strongly associated with highland plateaus (between 500 and 1200 m a.s.l.) [17]. Hence, this forest formation is subjected to a tropical climate in the northern part of their distribution and a subtropical climate in the southern part, with the latter consisting of four well-defined seasons [2]. Unlike tropical regions where the effects of microclimatic variation on insect assemblages are relatively well explored [18–21], this relationship is little known for subtropical forests.

Fruit-feeding butterflies are considered models for ecological studies and represent environmental quality and good predictors of diversity parameters, representing the diversity of other groups [1,22]. Their preference for a specific vegetational stratum along vertical gradients is relatively well described in the literature for the tropical zone worldwide [4,5,18,23–25]. These preferences are highly influenced by abiotic (temperature, humidity, luminosity, and wind speed) and biotic factors (vegetation architecture, resource availability, and presence of predators), as well as morphological traits related to flight performance [24,26,27]. Several studies indicate a clear segregation of strata leading to high beta diversity [25,28] and suggest species turnover as the main process that structures butterfly assemblages [18,29]. On the other hand, patterns in alpha diversity are inconsistent for fruit-feeding butterfly communities when comparing the understory and canopy. The most diverse stratum between these two strata depends on the region of the tropics considered (canopy for Central and South America butterflies [5,6,18,25,30] and understory for Afrotropical/Indo-Malaysian ones [4,31–33]).

This biogeographic incongruence highlights our lack of understanding of which factors are important in driving vertical stratification butterfly communities. It is known that canopies face more diel variation in conditions than the understory, thus having more stable conditions since the canopy itself would buffer the understory from insolation and air temperature changes throughout the day [34]. For ectothermic insects that respond to fine-scale abiotic factors, variation in microclimatic conditions can generate effects on species distribution comparable to the effect of the latitudinal gradient for other organisms [35]. Moreover, insect distributions may also be constrained at higher elevations as a result of reduced convective energy transfer and flight performance, and higher elevations may also lead to variation in adaptive strategies such as polymorphism and thermal tolerance [13,36], affecting the behavior and flight activity of butterflies [37]. However, it is unusual to find direct comparisons between butterfly diversity patterns and these potentially important proximal abiotic environmental factors, which would be the best prospect for solving our lack of knowledge on this topic.

Our study aims to evaluate how vertical stratification and microclimatic conditions affect the structure of fruit-feeding butterfly assemblages (alpha and beta diversity) in a subtropical Atlantic Forest region in southern Brazil. We aim to answer the following questions: (i) Is there an effect of temperature and humidity at different times of the day or night mediated by stratum on species richness and abundance? (ii) Is there a response of different components of beta diversity to environmental variation between strata? (iii) Can we find groups of species with preferences for one of the strata? We predict that the understory would show greater richness and abundance under warmer conditions, while the canopy would have greater diversity under wetter conditions. For beta diversity, we predict a difference in species composition between strata, in which canopy sites would

have less replacement of individuals of different species from site to site because species in this stratum can tolerate open habitat conditions and move between sites more freely than understory species.

## 2. Materials and Methods

The Floresta Nacional de São Francisco de Paula (FLONA-SFP) is located in the municipality of the same name, in the northeastern region of Rio Grande do Sul (centered at 29°25′22″ S; 50°23′11″ W), the southernmost Brazilian state, in South America (Figure 1). FLONA-SFP is located at approximately 900 m a.s.l. and comprises 1615 ha of native mixed ombrophilous forest with *Araucaria angustifolia* (Bertol.) Kuntze, as well as patches of *Pinus* sp. and *Eucalyptus* sp. plantations [38]. The vegetation is composed of elements from the dense ombrophilous forest (Atlantic Forest stricto sensu) and seasonal semideciduous forest, also exhibiting a mosaic landscape with subtropical highland grasslands [39]. The climate is subtropical without a dry season, with a mean annual rainfall of about 2000 mm and a mean temperature of 14.5 °C [40].

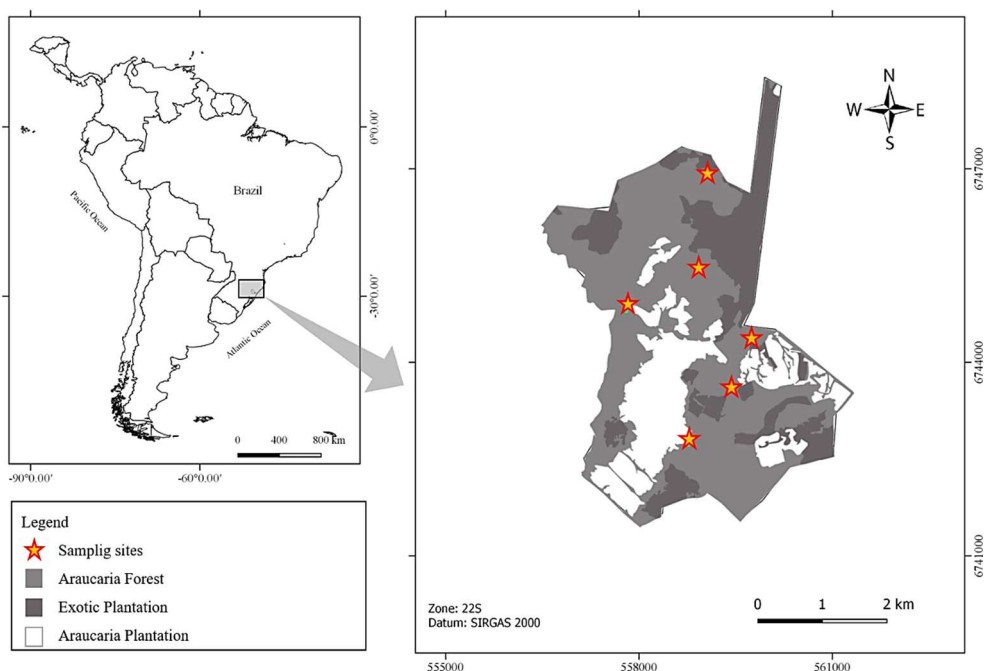

**Figure 1.** Map showing the location of the study area in the Rio Grande do Sul State, southern Brazil.

Samples were carried out with Van Someren–Rydon bait traps [41] in two periods, the first between November 2016 and March 2017, and the second between October 2017 and March 2018, representing the best period for butterfly sampling in the subtropical Atlantic Forest [2]. In both periods, data were collected monthly, and the traps remained open for about 8 days. We used six transects in Araucaria Forest, distant, at least 600 m from each other, with ten bait traps in each transect, totaling 60 traps. In each transect, five traps were placed in the canopy (~15 m above the ground, inside canopy tree crowns), alternating with five in the understory (1.5 m above the ground), at a horizontal distance of 20 m from each other. Each set of five bait traps per stratum composed a sampling unit. The baits consisted of a mixture of mashed banana and sugarcane juice, fermented for 48 h before sampling [41]. Traps were checked every 48 h, with baits replaced, and captured butterflies were identified in the field, also determining sex, marking with sequential Arabian numbers with a permanent pen, and then releasing all individuals. This mark–release procedure avoids the recounting of individuals during each sampling occasion. Exceptions were cases of notably difficult species identification or the need for vouchers (three specimens per species), which were collected for posterior identification. The voucher species are

deposited in the Lepidoptera Collection of Museu de Ciências Naturais Carlos Ritter, Instituto de Biologia at Universidade Federal de Pelotas (UFPel).

We measured the variation in air temperature and humidity between the canopy and understory using data loggers (Hobbo®) placed inside the traps in each stratum, in which the data loggers were exchanged among sampling units during the sampling occasions to cover all studied areas in FLONA-SFP. Measurements were carried out at 30 min intervals during all days within the sampling occasion. Months during which it was not possible to measure environmental variables were removed from the analysis, with seven out of 11 months of sampling remaining.

We consider each transect per stratum within a month as a sampling unit (2 stratum $\times$ 6 transects $\times$ 7 months = 84). Values for butterfly abundance, richness, mean temperature, and humidity represent the average values for each transect in each sampling month. We consider that transects sampled in the same month were not independent, adjusting for this in models for alpha and beta diversity.

We explored the variation in diversity profile between the canopy and understory using rarefaction and extrapolation curves based on Hill number and distinct *q* parameters [42,43]. These parameters measure the sensitivity to increments in abundance weight, where $q = 0$ is analogous to species richness, and $q = 1$ and $q = 2$ are analogous to the Shannon index (evenness) and Simpson index (dominance), respectively [43].

We investigated whether the mean and standard deviation of temperature and humidity affected the richness or abundance of fruit-feeding butterflies between strata and the period of the day (day or night). We considered day periods between 8:00 and 18:00, and night periods between 18:00 and 08:00, defined according to the predicted daily activity for fruit-feeding butterflies [44]. We employed a generalized mixed model to test the following hypothesis:

$$S_i \text{ or } N_i = \alpha_{j(i)} + \beta 1_i \times Tmean_{day} + \beta 2_i \times Tsd_{night} + \beta 3_i \times Tmean_{day}: Strata_{und} + \beta 4_i \times Tsd_{night}:$$
$$Strata_{und} + \beta 5_i \times Hmean_{day}:Strata_{can} + \beta 6_i \times Hmean_{day}: Strata_{und} + \varepsilon,$$

along with the following structure of the random effects for intercepts:

$$\alpha_j \sim Normal(\mu_\alpha, \sigma_\alpha),$$

where S and N are the richness and abundance, respectively, for each transect in each sampling month, *α* is the intercept (canopy by default), *β*1 to *β*6 are the slopes, the index *i* indicates the sampling units, and *j* represents the random term associated with the sampling month that surveys were realized. Strata is a categorical variable with two levels (can = canopy or und = understory), *Tmean* and *Tsd* are the mean and standard deviation of temperature, and *Hmean* is the mean humidity, subscribed by the period of day (day or night). *ε* is the residual that we assumed to come from a Poisson distribution for S and a negative binomial distribution for N. We evaluated the collinearity among predictors by employing a variance inflator factor (VIF) [45], and the predictor variables were standardized (centered at zero and scaled).

To analyze butterfly assemblage species composition, we accessed the total beta diversity among sites (βtot) and decomposed it into the relativized species replacement component (βrep) and the richness differences component (βric). We generated the beta diversity components using abundance data and the Sørensen index of dissimilarity. These components were used in a permutational multivariate analysis of variance to evaluate the effects of environmental variables (stratum, mean temperature of the day, the standard deviation of temperature at night, and mean humidity of the day) over species composition. We also tested the homogeneity in group variation [46] and constraint permutation within the sampling month.

We also used indicator value analysis (IndVal) [47] to check whether species or groups of species could be considered indicators of canopy and understory habitats. This analysis uses the relationship between species abundances and environment specificity and fidelity.

Values of specificity equal to 1 indicate that per species were present in all sites and sampling occasions, while values of fidelity equal to 1 indicate the probability of finding the same species in the same group of sites in all samplings [48].

All data analysis was performed in the R environment (v. 4.1.1) [49]. For estimates of sampling coverage and diversity profiles (function *ChaoHill* described in [50], we used package iNEXT (v. 2.0.20)) [51]. Mixed models were performed with the *lme4* package (v. 1.1-27.1) [52], and the variance inflator factor was performed with the *car* package (v. 3.0-11) [53]. IndVal was assessed with the function *multipatt* of the *indicspecies* package (v. 1.7.9) [54]. For beta diversity analysis, we used the packages *vegan* (v. 2.5-7) [55] and *BAT* (v. 2.9.2) [56]. All graphics were generated with the package *ggplot2* (v. 3.3.5) [57]. All code needed to perform the analysis used for this manuscript can be found at https://github.com/richterbine/Vertical_Stratification_bfly/tree/master, accessed on 3 April 2023.

## 3. Results

After 2450 trap days of sampling effort, we captured 1347 individuals representing 38 species of fruit-feeding butterflies distributed in the four subfamilies of Nymphalidae (Table S1, supporting information). Canopy had 30 species and 563 individuals, whereas understory had 25 species and 784 individuals. Both strata exhibited 99% sampling coverage, indicating an adequate representation of butterfly assemblages. Seventeen species were shared between strata, 13 were exclusive to the canopy, and eight were exclusive to the understory. Satyrinae was the most represented subfamily with 22 species (83.7%), followed by Charaxinae with eight (12.4%), Biblidinae with seven (3.5%), and Nymphalinae with only one species (0.4%).

The rarefaction and extrapolation diversity curves (Supplementary Figure S1), as well as the diversity profiles (Figure 2), indicated an inversion of the most diverse strata according to the weight given to abundance. At $q = 0$ and $q = 1$, the strata did not differ in diversity, whereas, at $q = 2$, the understory had lower dominance than the canopy. This indicates that species abundance in the understory is more evenly distributed, mainly because the canopy exhibited a high dominance by a single species *Carminda paeon* (Godart, [1824]), which made up 53% of the total abundance of the stratum (Table S1, supporting information).

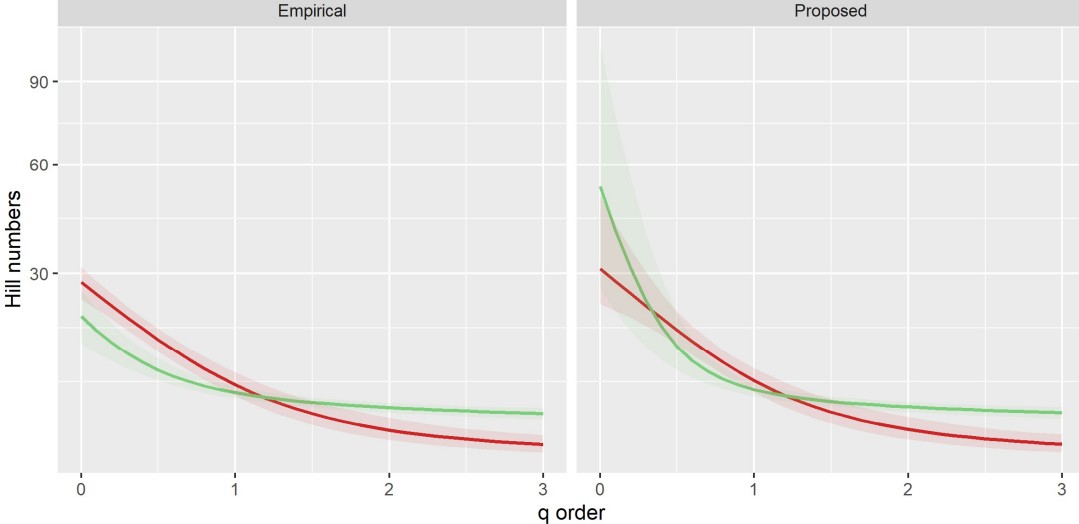

**Figure 2.** Empirical and proposed diversity profiles based on the *q* statistic for fruit-feeding butterfly assemblages for the canopy (red line) and understory (green line) sampled in mixed ombrophilous forest in southern Brazil during November 2016–March 2017 and October 2017–March 2018. Confidence intervals (95%) were constructed on the basis of bootstrap resampling (1000 iterations).

### 3.1. Environmental Effects on Richness and Abundance

The effect of environmental variables was only significant for abundance since, for richness, the model was worse than the null model (Table 1). The abundance of fruit-feeding butterflies increased with temperature and relative humidity of the day independently of the stratum (no interaction, Figure 3a,c), while the effect of variation in temperature at night was mediated by the strata (Table 2). The canopy showed a larger increase in predicted abundance when the variation in temperature increased (Figure 3b), whereas, for the understory, the increase in abundance was less pronounced.

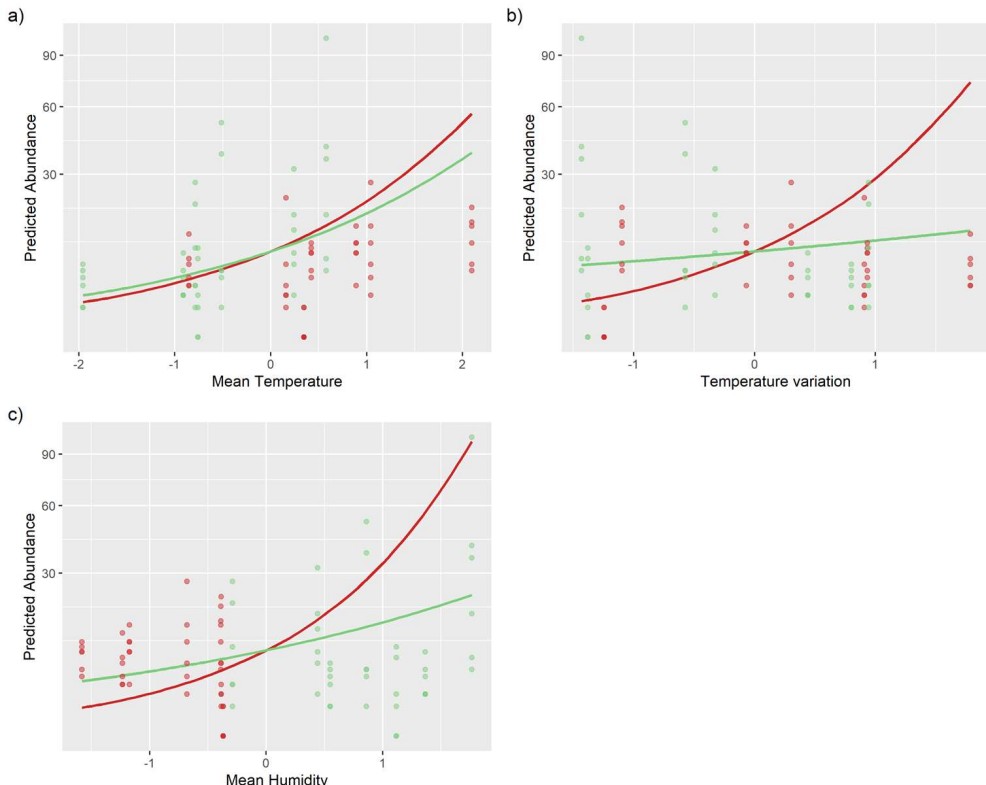

**Figure 3.** Effects of environmental variables that explained the abundance patterns of fruit-feeding butterfly assemblages sampled in mixed ombrophilous forest in southern Brazil during November 2016–March 2017 and October 2017–March 2018. Red colors indicate sites and regression lines related to the canopy, while green colors indicate sites and regression lines related to the understory. (**a**) Effects of mean temperature at day, highlighting the lack of interaction with the stratum. (**b**) Effects of the standard deviation of temperature at night and their dependency on the stratum. (**c**) Effects of the mean humidity at day on abundance, showing the positive relationship with the canopy. The y-axis was square-root-transformed for better visualization of the patterns. The predictor variables were scaled before performing the analysis to have a mean equal to 0 and a standard deviation equal to 1.

**Table 1.** Model selection table for models for richness and abundance of fruit-feeding butterfly assemblages sampled in mixed ombrophilous forest in southern Brazil during November 2016–March 2017 and October 2017–March 2018. In the null model, the response variables were modeled by 1, whereas, in the full model, they were modeled considering the environmental variables. We considered the best model to have the lowest AICc and a delta < 2. [1] Pseudo-$R^2$ calculated using the trigamma method and considering both fixed and random effects (conditional).

| | df | logLik | AICc | Delta | Weight | Pseudo-$R^2$ |
|---|---|---|---|---|---|---|
| Model for richness | | | | | | |
| Null (Poisson) | 2 | −163.125 | 330.400 | 0.000 | 0.985 | |
| Full model (Poisson) | 8 | −160.424 | 338.800 | 8.370 | 0.015 | 0.417 [1] |
| Model for abundance | | | | | | |
| Full model (negative binomial) | 9 | −258.573 | 537.600 | 0.000 | 0.908 | 0.472 [1] |
| Null (negative binomial) | 3 | −267.928 | 542.200 | 4.580 | 0.092 | |

**Table 2.** Effect of environmental variables in the abundance of fruit-feeding butterfly assemblages of sampled in mixed ombrophilous forest in southern Brazil during November 2016–March 2017 and October 2017–March 2018, according to the model selection. The values of parameters estimated are in the log-link scale. SE—standard error. Bold values indicate the terms that were significant at an alpha threshold of 0.05.

| | Estimate | SE | z-Value | Pr(>\|z\|) |
|---|---|---|---|---|
| Intercept | 2.120 | 0.333 | 6.372 | 0.000 |
| Mean temperature (day) | 0.913 | 0.429 | 2.127 | **0.033** |
| Temperature variation (night) | 1.221 | 0.457 | 2.674 | **0.007** |
| Mean temperature (day): understory | −0.183 | 0.388 | −0.471 | 0.638 |
| Temperature variation (night): understory | −0.979 | 0.416 | −2.352 | **0.019** |
| Canopy: mean humidity (day) | 1.400 | 0.464 | 3.016 | **0.003** |
| Understory: mean humidity (day) | 0.564 | 0.420 | 1.343 | 0.179 |

*3.2. Environmental Effects on Beta Diversity*

The total beta diversity (βtot) in the subtropical Araucaria Forest was 0.885, driven by a small difference between the βrep and βric components (0.420, sd = 0.278 and 0.465, sd = 0.271, respectively). We observed that βtot was affected only by strata (Table 3), despite the lower $R^2$ value (0.03), whereby the variation between understory sites was higher than the variation between canopy sites (Figure 4a,d). The beta components βrep and βric (Figure 4b,c) did not show effects of environmental variables evaluated or strata, although we observed that the stratum had a similar $R^2$ value to that found in the model for βtot (Table 3). Moreover, IndVal generated two groups, with 11 out of the 38 species associated with one or the other stratum (Table 4). Five species belonging to the Satyrini tribe (Satyrinae), Charaxinae, Biblidinae, and Nymphalinae, can be considered canopy indicators, and six species corresponding to Satyrinae can be considered understory indicators.

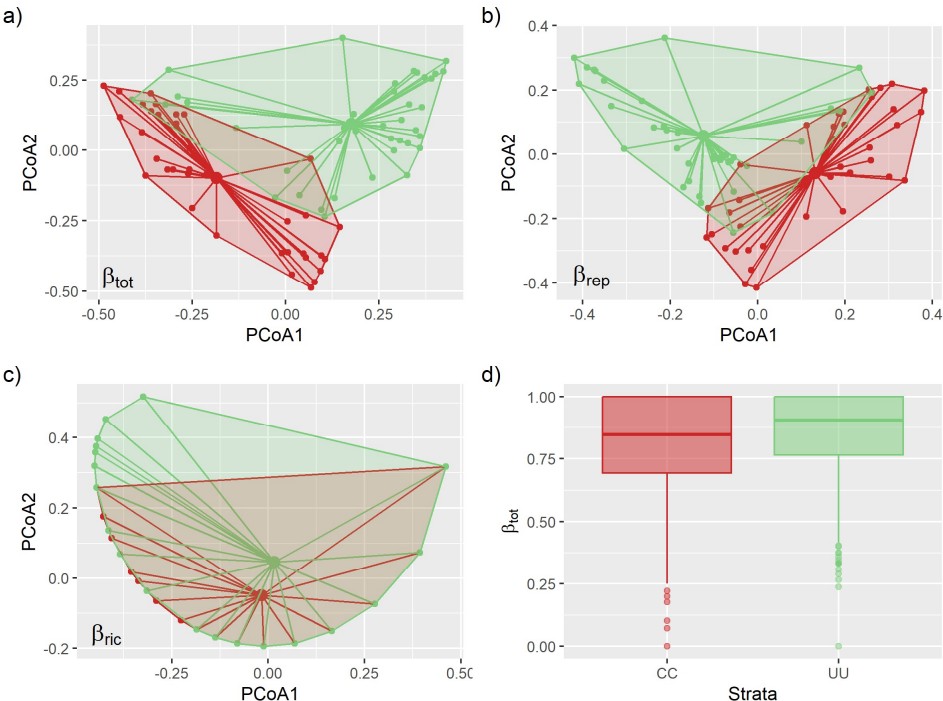

**Figure 4.** Beta diversity components and their relationship with environmental variables and the stratum of fruit-feeding butterfly assemblages sampled in mixed ombrophilous forest in southern Brazil during November 2016–March 2017 and October 2017–March 2018. (**a–c**) The dispersion of points related to strata for the component of total dissimilarity ($\beta$tot), the component of abundance replacement ($\beta$rep), and the component of abundance differences ($\beta$ric), respectively. (**d**) Boxplot showing the effects of strata in $\beta$tot, indicating a larger dissimilarity among understory sites (UU, green color) than among canopy sites (CC, red color).

**Table 3.** Effects of environmental variables on partitioning beta diversity dissimilarity in fruit-feeding butterfly assemblages sampled in mixed ombrophilous forest in southern Brazil during November 2016–March 2017 and October 2017–March 2018. $\beta$tot—component related to total dissimilarity among pairwise sites; $\beta$rep—component related to abundance replacement among sites; $\beta$ric—component of abundance difference due to the gain/loss of individuals among sites.; df—degrees of freedom; SumOfSqs—sum of squares. Bold values indicate the significant terms at a *p* threshold of 0.05.

| | **Df** | **SumOfSqs** | **R$^2$** | **F** | ***p*** |
|---|---|---|---|---|---|
| $\beta_{tot}$ | | | | | |
| Mean temperature (day) | 1 | 0.562 | 0.018 | 1.625 | 0.081 |
| Temperature variation (night) | 1 | 0.378 | 0.012 | 1.094 | 0.970 |
| Mean humidity (day) | 1 | 0.561 | 0.018 | 1.623 | 0.161 |
| Strata | 1 | 1.165 | 0.037 | 3.367 | **0.001** |
| Residual | 73 | 25.246 | 0.811 | | |
| Total | 77 | 31.144 | 1.000 | | |
| $\beta_{rep}$ | | | | | |
| Mean temperature (day) | 1 | −0.493 | −0.050 | −4.265 | 0.983 |
| Temperature variation (night) | 1 | −0.072 | −0.007 | −0.619 | 0.999 |
| Mean humidity (day) | 1 | 0.074 | 0.008 | 0.639 | 0.779 |
| Strata | 1 | 0.339 | 0.035 | 2.937 | 0.060 |
| Residual | 73 | 8.435 | 0.864 | | |
| Total | 77 | 9.766 | 1.000 | | |

**Table 3.** *Cont.*

|  | Df | SumOfSqs | R$^2$ | F | *p* |
|---|---|---|---|---|---|
| $\beta_{ric}$ |  |  |  |  |  |
| Mean temperature (day) | 1 | 1.035 | 0.093 | 8.357 | 0.143 |
| Temperature variation (night) | 1 | 0.143 | 0.013 | 1.156 | 0.214 |
| Mean humidity (day) | 1 | 0.156 | 0.014 | 1.262 | 0.228 |
| Strata | 1 | 0.558 | 0.050 | 4.509 | 0.485 |
| Residual | 73 | 9.040 | 0.810 |  |  |
| Total | 77 | 11.155 | 1.000 |  |  |

**Table 4.** Indicator species analysis (IndVal), showing the values of fidelity (A) and specificity (B) for fruit-feeding butterflies between the canopy and understory sampled in a mixed ombrophilous forest in southern Brazil, during November 2016–March 2017 and October 2017–March 2018. Names in the parentheses indicate the subfamily and tribe, respectively. Sat—Satyrinae, Cha—Charaxinae, Bib—Biblidinae, Nym—Nymphalinae. Only species with *p* values < 0.05 are shown.

|  | A | B | Stat | *p* |
|---|---|---|---|---|
| Canopy |  |  |  |  |
| *Carminda paeon* (Sat, Satyrini) | 0.906 | 0.632 | 0.756 | 0.001 |
| *Zaretis strigosus* (Cha, Anaeini) | 1.000 | 0.368 | 0.607 | 0.001 |
| *Epiphile orea* (Bib, Epiphilini) | 0.944 | 0.290 | 0.523 | 0.001 |
| *Memphis moruus* (Cha, Anaeini) | 1.000 | 0.263 | 0.513 | 0.001 |
| *Smyrna blomfildia* (Nym, Coeini) | 1.000 | 0.105 | 0.324 | 0.050 |
| Understory |  |  |  |  |
| *Pseudodebis ypthima* (Sat, Satyrini) | 1.000 | 0.575 | 0.758 | 0.001 |
| *Eryphanis reevesii* (Sat, Brassolini) | 0.985 | 0.500 | 0.702 | 0.001 |
| *Forsterinaria quantius* (Sat, Satyrini) | 0.988 | 0.325 | 0.567 | 0.001 |
| *Morpho epistrophus* (Sat, Morphini) | 0.979 | 0.250 | 0.495 | 0.003 |
| *Opoptera fruhstorferi* (Sat, Brassolini) | 1.000 | 0.225 | 0.474 | 0.007 |
| *Caligo martia* (Sat, Brassolini) | 1.000 | 0.150 | 0.387 | 0.023 |

## 4. Discussion

In this study, we report for the first time the response of fruit-feeding butterflies to environmental conditions across vertical strata in a subtropical region, highlighting the role of daily variation in microclimatic conditions on the structure of these assemblages. We emphasize the importance of the mean and deviation of temperature for abundance, not only for the period of butterfly activity but also for the nighttime, especially in canopy communities. In this stratum, fluctuation throughout the day is more pronounced than in the understory since, during the day, the canopy reaches higher average temperature values than the understory [58], whereas, at night, the temperature is homogenized between the strata [12], which was also confirmed by our data (Supplementary Figure S2). Although these organisms have a diurnal or crepuscular activity [44], they need adaptations to survive at night, especially in environments such as the Araucaria forests, which are associated with higher altitudes and where the observed minimum temperatures can reach below 10 °C in the summer period.

It is recognized that abiotic conditions, such as temperature, rainfall, and luminosity, affect the richness and abundance of fruit-feeding butterflies locally [18,19,21]. Thus, thermal variation in the microhabitat scale and the relationship with physiological and morphological components may be important for understanding how species and individuals are distributed along environmental gradients [35,59]. On the other hand, at a macroecological scale, these abiotic factors can determine the seasonal pattern in species occurrence in subtropical and temperate zones, limiting the growing season to the warmer months due to physiological constraints [59–62]. In temperate regions, climatic patterns are the most important factors influencing the distribution of species richness [19,61] and may override the effect of microclimatic variation. We observed that richness did not



respond to daily variation or differ between strata. We suggest that the lack of stratification in richness in our study could have been an effect of macroclimatic conditions, which can determine the temperature homogenization at night, as well as influence the lower structural complexity of the forest (a relatively lower canopy height of about 10 to 25 m when compared with the Amazonian Forest, for example). Considering a microscale, the distribution of host plants can contribute to this lack of differentiation in richness, creating vertical connections between the understory and canopy using host plants in the immature stages [63], in which females of butterflies using the canopy may lay eggs in the low and middle understory. In this way, some canopy-dweller species can be caught in understory traps, thus inflating species richness.

We demonstrate that daily microclimatic variations are important in defining abundance, having distinct effects depending on the vertical stratum. Even though the canopy is warmer and drier than the understory (Supplementary Figure S2), an increase in mean temperature and humidity leads to an increase in abundance independently of strata despite the higher abundance of butterflies in the canopy. However, the vertical stratification on abundance was only observed when considering the variation of microclimatic conditions; an increase in the temperature variation had a positive effect on abundance for the canopy. This result highlights that the amplitude of variation in the temperature and humidity is important for species that inhabit the canopy layer. On the other hand, the butterflies that inhabit the understory have more stability concerning the microclimatic conditions. Unstable environments would allow individuals with different ecological requirements to be able to find the bait trap when the environment is within their optimal temperature range of activity [26]. This may be related to the difference in the intensity of the effect of environmental variation on the abundance in the canopy, corroborating the findings of Checa et al. [21] in warmer tropical forests.

The species composition is highly dissimilar between canopy and understory, as demonstrated in several studies regarding vertical stratification for fruit-feeding butterflies [6,18,21,25,30]. Our study corroborated this pattern and demonstrated that the understory has more dissimilarity among sites than the canopy. However, considering the components of beta diversity, the variables evaluated do not seem responsible for determining the processes of replacement and loss/gain of individuals within the community. The lack of effect of climatic conditions may be related to the scale of the study (microclimatic variations), and the greater dissimilarity of the understory may be due to the substitution in the number of individuals among the species. This variation is, possibly, associated with other characteristics across this gradient of vertical stratification, such as dispersal limitations of butterflies in the understory as a result of their specialization for inhabiting closed forests, or they may be determined by variables that act on a larger scale [64]. Furthermore, other facets of diversity, e.g., functional or phylogenetic, can influence the patterns of species distributions [64–66] and should be considered to better understand how assemblages are structured.

Through the analysis of indicator species, it is possible to define which species are more related to each stratum. Similar to other studies [18,25,26], we observed the strong association of Satyrinae with the understory, while the canopy encompassed species of the other subfamilies [18,25,26]. The association between Satyrinae and understory may result from a phylogenetic constraint [28] since the evolution of the group supposedly took place in closed forest environments [67]. The tribes Brassolini and Morphini have adaptations to use the understory of forests, such as larger body sizes and darker colors [26,35] and a short period of optimum adult activity during the year [62,68]. On the other hand, the more heterogenous canopy is defined by species with high mobility and the ability to tolerate greater environmental variations, such as many Charaxinae and Nymphalinae that are strongly associated with sunny habitats [18,26] and have fast thermoregulation with strong bodies and rapid flights [24,69]. This result may help us to understand the patterns observed for beta diversity. The highly mobile butterflies found in the canopy can lead to a homogenization of composition because they use the canopy as a corridor (or,

more precisely, a sheet) to disperse between open patches perceiving this habitat as a green carpet [5,70].

From studies performed in the Neotropical region, diversity patterns of fruit-feeding butterflies' distribution between canopy and understory may respond to several factors. Among them, the main one is the climate, which influences forest structure and seasonality, directly affecting fruit-feeding butterflies [60]. However, we demonstrate that microclimatic variation is very important to describe fruit-feeding butterfly assemblages' local structure and abundance patterns. Our study can help to elucidate the distinct patterns found in this guild, mainly for alpha diversity in studies concerning vertical stratification. It seems that, for beta diversity, the pattern is more marked and consistent, with great convergence in all studies, indicating an intense selection pressure filtering the butterfly species between the canopy and understory. This study was the first to evaluate vertical stratification for fruit-feeding butterflies in a subtropical forest, helping to elucidate mechanisms and processes influencing these insect assemblages in the Neotropics. We hope that further studies strive to take microclimatic variation into account for short- and long-term evaluations to improve our knowledge of how assemblages are structured in subtropical areas worldwide.

**Supplementary Materials:** The following supporting information can be downloaded at https://www.mdpi.com/article/10.3390/d15040560/s1: Figure S1: Rarefaction and extrapolation curves for fruit-feeding butterflies species diversity based on the Hill numbers for the canopy and the understory; Figure S2: Relationship between environmental variables and vertical stratum considering the day and night periods; Table S1: Species list of fruit-feeding butterflies in a mixed ombrophilous forest; Table S2: Relationship between environmental variables and vertical stratum considering the day and night periods.

**Author Contributions:** Conceptualization, A.R., M.d.S.M.J. and C.A.I.; methodology, A.R. and C.A.I.; validation, A.R., C.A.I. and M.d.S.M.J.; formal analysis, A.R.; investigation, A.R., K.G. and C.A.I.; resources, A.R., K.G. and C.A.I.; data curation, A.R., K.G. and C.A.I.; writing—original draft preparation, A.R.; writing—review and editing, A.R., M.d.S.M.J. and C.A.I.; supervision, C.A.I.; project administration, A.R., M.d.S.M.J., K.G. and C.A.I.; funding acquisition, A.R. and C.A.I. All authors have read and agreed to the published version of the manuscript.

**Funding:** This research was funded by Coordenação de Aperfeiçoamento de Pessoal de Nível Superior (CAPES) for graduate fellowships (financial code 001) to A.R. and K.G.; CNPq for a productivity scholarship (309616/2015-8) for M.M.J.; National Institutes for Science and Technology (INCT) in Ecology, Evolution, and Biodiversity Conservation, supported by MCTIC/CNPq (proc. 465610/2014-5) and FAPEG (proc. 201810267000023) for A.R. and C.A.I.

**Institutional Review Board Statement:** Not applicable.

**Data Availability Statement:** All code needed to perform the analysis used for this manuscript can be found at https://github.com/richterbine/Vertical_Stratification_bfly/tree/master (accessed in 3 April 2023) and in the Supplementary Materials.

**Acknowledgments:** The authors thank colleagues in the Laboratório de Ecologia de Lepidoptera and other friends that helped in field expeditions. The authors are grateful to the staff of Floresta Nacional de São Francisco de Paula for providing logistic assistance during the sampling periods, to Dra Karen Mustin for revision of the language of this manuscript, to the two anonymous reviewers for critically reviewing and valuable contributions to the manuscript, and to André Victor Lucci Freitas for help in the identification of some butterflies. Samples were procured under ICMBio permanent license number 45673-1 and research license numbers 54298-1 and 59568-1. This publication is part of the RedeLep "Rede Nacional de Pesquisa e Conservação de Lepidópteros" (National Network for Research and Conservation of Lepidoptera).

**Conflicts of Interest:** The authors declare no conflict of interest.

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
