# Peer review of "Microclimatic Fluctuation throughout the Day Influences Subtropical Fruit-Feeding Butterfly Assemblages between the Canopy and Understory"

_diversity, doi:10.3390/d15040560_

Round 1

Reviewer 1 Report

Comments to authors:

1.       In the introduction add some text about Vertical connectivity. Indeed, the different layers of the forest are connected by vertical pathways, such as vines, lianas, and epiphytes. These pathways provide important routes for insect movement and dispersal, which can help to maintain insect diversity across the forest.

2.       In lines 142-146 you are indeed mentioning the use of Hill Numbers. Please add mention to Hill Series and cite a relevant paper like

Chao, A., Chiu, C. H., & Jost, L. (2014). Unifying species diversity, phylogenetic diversity, functional diversity, and related similarity and differentiation measures through Hill numbers. Annual review of ecology, evolution, and systematics45, 297-32

3.       Concerning the use of Beta Partition I think that the approach of Baselga is not adequate since ~in your study you are not interested in “nestedness”

The Carvalho et al (2013) approach, partitions beta diversity into two components: turnover (due to species replacement) and richness differences (due to differences in species richness between sites). This approach provides a more ecologically meaningful way of understanding beta diversity.

Carvalho, J.C., Cardoso, P., Borges, P.A.V., Schmera, D. & Podani, J. (2013). Measuring fractions of beta diversity and their relationships to nestedness: a theoretical and empirical comparison of novel approaches. Oikos, 122: 825–834. DOI:10.1111/j.1600-0706.2012.20980.x

So, in the context of beta diversity partitioning, using nestedness as a component measures the degree to which small, isolated habitats differ from larger, connected habitats in terms of their species composition, while using richness differences as a component measures the overall difference in species richness between two sites, regardless of their compositional similarity or differences. Both nestedness and richness differences can be useful components of beta diversity partitioning, but they provide different information about the underlying ecological processes driving beta diversity.

See the R Package BAT

Cardoso, P., Rigal, F., & Carvalho, J. C. (2015). BAT–Biodiversity Assessment Tools, an R package for the measurement and estimation of alpha and beta taxon, phylogenetic and functional diversity. Methods in Ecology and Evolution6(2), 232-236.

Author Response

We greatly appreciated the comments by the two referees on our manuscript entitled “Microclimatic fluctuation throughout the day influences sub-tropical fruit-feeding butterfly assemblages between the canopy and understory”, which was submitted for publication in Diversity (Diversity-2304171).

We have taken into account all the comments made by the reviewers while preparing this new version of the ms for resubmission. We are most grateful for the valuable criticisms on the first version of the ms – we do think that this revised version is considerably improved, and hope that you find our study appropriate for Diversity.

Please see below the major comments from both reviewers, followed by our response to each of them.

Sincerely, Cristiano A. Iserhard

Comments and responses

1 - In the introduction add some text about Vertical connectivity. Indeed, the different layers of the forest are connected by vertical pathways, such as vines, lianas, and epiphytes. These pathways provide important routes for insect movement and dispersal, which can help to maintain insect diversity across the forest.

Response 1: Thank you for the comment. We will add this point of view but consider the distribution of the host plant along the vertical gradient as the trigger for movement between forest layers (Queiroz, 2002). Also, we opted to include this information in the discussion section to corroborate that host plant distribution along a vertical gradient can lead to non-differentiation in richness among forest layers, mainly because some species that preferentially use the canopy can be caught in understorey traps when they are searching for their host plant.

2 - In lines 142-146 you are indeed mentioning the use of Hill Numbers. Please add mention to Hill Series and cite a relevant paper like

Chao, A., Chiu, C. H., & Jost, L. (2014). Unifying species diversity, phylogenetic diversity, functional diversity, and related similarity and differentiation measures through Hill numbers. Annual review of ecology, evolution, and systematics, 45, 297-32

Response 2: Done.

3 - Concerning the use of Beta Partition I think that the approach of Baselga is not adequate since ~in your study you are not interested in “nestedness”

The Carvalho et al (2013) approach, partitions beta diversity into two components: turnover (due to species replacement) and richness differences (due to differences in species richness between sites). This approach provides a more ecologically meaningful way of understanding beta diversity.

Carvalho, J.C., Cardoso, P., Borges, P.A.V., Schmera, D. & Podani, J. (2013). Measuring fractions of beta diversity and their relationships to nestedness: a theoretical and empirical comparison of novel approaches. Oikos, 122: 825–834. DOI:10.1111/j.1600-0706.2012.20980.x

So, in the context of beta diversity partitioning, using nestedness as a component measures the degree to which small, isolated habitats differ from larger, connected habitats in terms of their species composition, while using richness differences as a component measures the overall difference in species richness between two sites, regardless of their compositional similarity or differences. Both nestedness and richness differences can be useful components of beta diversity partitioning, but they provide different information about the underlying ecological processes driving beta diversity.

See the R Package BAT

Cardoso, P., Rigal, F., & Carvalho, J. C. (2015). BAT–Biodiversity Assessment Tools, an R package for the measurement and estimation of alpha and beta taxon, phylogenetic and functional diversity. Methods in Ecology and Evolution, 6(2), 232-236.

Response 3: We agree with the reviewer and redo the analysis using the suggested methodology. The new analysis, results and discussion have been added to the text.

TEXT MODIFICATIONS:

The main changes have been highlighted in red in the text, and below is a summary of the changes in the topics:

  1. Abstract – We have improved the abstract section and rewritten the results for the beta diversity partitioning and the discussion about them.
  2. Introduction – We have corrected the spelling throughout the text and rewritten the predictions for the key questions.
  3. Material and Methods –We have corrected spelling throughout the text and rewritten parts of the data analysis section as suggested by the reviewers. We add text and a new reference. As suggested by reviewer 1, we changed the approach to evaluating beta diversity partitioning by now using the methodology proposed independently by Podani and Schmera (2011) and Carvalho et al. (2012). The data were evaluated using the BAT package (Cardoso et al. 2015).
  4. Results - We have corrected spelling throughout the text and rewritten the results about beta diversity partitioning. Since the results for beta diversity have changed, we have updated Table 3 and Figure 4 with the new values. As suggested by reviewer 2, we have changed the terms in the tables from acronyms to variable names in Tables 1, 2, and 3.
  5. Discussion – Following the recommendations of reviewer 1, we add some discussion about the role of vertical pathways connecting the understory and canopy. We change the discussion and some references about beta diversity partitioning. We have corrected the spelling throughout the text.
  6. Authors Contributions – We add this section in the main text, following the “Supplementary Materials” section.
  7. References – we updated the references used in this manuscript.

Reviewer 2 Report

The authors studied fruit-feeding butterfly assemblages in south Brazil. They have used standard methods, i.e. bait traps, in two strata, forest understory and canopy, but the authors also used data loggers and measured temperature and humidity. Using mixed models as a tool for analyses, the authors found that whereas the two strata did not differ in diversity, the understory has lower dominance than the canopy but also a higher species turnover. The abundance of butterflies increased with temperature and humidity regardless of the stratum, however, the strata differed in temperature variation at night.

The study is well planned, the results are clear and the paper is well written. However, it is not easy to read it as there are too much of abbreviations with sometimes confusing explanations. For instance, dBC stands for the "partition of beta diversity into total dissimilarity among sites" (line 172), "total beta diversity" (line 231), "component related with total Bray-Curtis index of dissimilarity among pairwise sites" (line 263), or "total dissimilarity"(line 291-292). I can imagine that it is all of these together, but it needs to be united and especially if it is calculated using Bray-Curtis index, this should be mentioned in the Methods and not in a figure legend. Similarly, the authors should use in tables rather than something like "ric.pois.null" or "Tmean.day" the real variable names. Otherwise, the study is very interesting and deserves to be published. 

Author Response

We greatly appreciated the comments by the two referees on our manuscript entitled “Microclimatic fluctuation throughout the day influences sub-tropical fruit-feeding butterfly assemblages between the canopy and understory”, which was submitted for publication in Diversity (Diversity-2304171).

We have taken into account all the comments made by the reviewers while preparing this new version of the ms for resubmission. We are most grateful for the valuable criticisms on the first version of the ms – we do think that this revised version is considerably improved, and hope that you find our study appropriate for Diversity.

Please see below the major comments from both reviewers, followed by our response to each of them.

Sincerely, Cristiano A. Iserhard

Comments and responses

Response 1: Thank you very much for your comments. We revised the text to make it more fluid, standardized the acronyms, and changed the variable names in the tables.

TEXT MODIFICATIONS:

The main changes have been highlighted in red in the text, and below is a summary of the changes in the topics:

  1. Abstract – We have improved the abstract section and rewritten the results for the beta diversity partitioning and the discussion about them.
  2. Introduction – We have corrected the spelling throughout the text and rewritten the predictions for the key questions.
  3. Material and Methods –We have corrected spelling throughout the text and rewritten parts of the data analysis section as suggested by the reviewers. We add text and a new reference. As suggested by reviewer 1, we changed the approach to evaluating beta diversity partitioning by now using the methodology proposed independently by Podani and Schmera (2011) and Carvalho et al. (2012). The data were evaluated using the BAT package (Cardoso et al. 2015).
  4. Results - We have corrected spelling throughout the text and rewritten the results about beta diversity partitioning. Since the results for beta diversity have changed, we have updated Table 3 and Figure 4 with the new values. As suggested by reviewer 2, we have changed the terms in the tables from acronyms to variable names in Tables 1, 2, and 3.
  5. Discussion – Following the recommendations of reviewer 1, we add some discussion about the role of vertical pathways connecting the understory and canopy. We change the discussion and some references about beta diversity partitioning. We have corrected the spelling throughout the text.
  6. Authors Contributions – We add this section in the main text, following the “Supplementary Materials” section.
  7. References – we updated the references used in this manuscript.

Round 2

Reviewer 1 Report

Many thanks for having performed the requested changes I have suggested. Now the manuscript can be accepted.